A new specimen of the Early Cretaceous bird Hongshanornis longicresta: insights into the aerodynamics and diet of a basal ornithuromorph

Chiappe Luis M. 1 chiappe@nhm.org
Zhao Bo 2
O’Connor Jingmai K. 3
Chunling Gao 2
Wang Xuri 2 4
Habib Michael 5
Marugan-Lobon Jesus 6
Meng Qingjin 7
Cheng Xiaodong 2
1 Dinosaur Institute, Natural History Museum of Los Angeles County , Los Angeles, CA , USA
2 Dalian Natural History Museum , District Dalian , PR China
3 Institute of Vertebrate Paleontology and Paleoanthroplogy , Beijing , PR China
4 Institute of Geology, Chinese Academy of Geological Sciences , Beijing , PR China
5 University of Southern California, Health Sciences Campus , Los Angeles, CA , USA
6 Unidad de Paleontología, Dpto. Biología, Universidad Autónoma de Madrid , Cantoblanco , Spain
7 Beijing Natural History Museum , Beijing , PR China
Thewissen J
Electronic publication date: 2014 Jan 2
Publication date: 2014
Volume: 2
Electronic Location ID: e234
Received 2013 Aug 30; Accepted 2013 Dec 11
Copyright: © 2014 Chiappe et al.
Copyright year: 2014
Copyright holder: Chiappe et al.
License: This is an open access article distributed under the terms of the Creative Commons Attribution License, which permits unrestricted use, distribution, and reproduction in any medium, provided the original author and source are credited.
License URL: https://creativecommons.org/licenses/by/3.0/

Keywords: Cretaceous, Birds, Aerodynamics, Morphology, China

Funding: Doreen and Glenn Gee to the Dinosaur Institute This research was supported by the “Study of Morphology and Systematics of the Jehol Biota” project of the “Building an Innovative Team Plan” program of the Chinese government, and by donations from Doreen and Glenn Gee to the Dinosaur Institute. The funders had no role in study design, data collection and analysis, decision to publish, or preparation of the manuscript.

==============================
The discovery of Hongshanornis longicresta, a small ornithuromorph bird with unusually long hindlimb proportions, was followed by the discovery of two closely related species, Longicrusavis houi and Parahongshanornis chaoyangensis. Together forming the Hongshanornithidae, these species reveal important information about the early diversity and morphological specialization of ornithuromorphs, the clade that contains all living birds. Here we report on a new specimen (DNHM D2945/6) referable to Hongshanornis longicresta that contributes significant information to better understand the morphology, trophic ecology, and aerodynamics of this species, as well as the taxonomy of the Hongshanornithidae. Most notable are the well-preserved wings and feathered tail of DNHM D2945/6, which afford an accurate reconstruction of aerodynamic parameters indicating that as early as 125 million years ago, basal ornithuromorphs had evolved aerodynamic surfaces comparable in size and design to those of many modern birds, and flight modes alike to those of some small living birds.

Introduction

Until the recent discoveries from the Jehol Group (Zhou & Zhang, 2007; Chiappe, 2007; O’Connor, Chiappe & Bell, 2011; Zhou, Zhou & O’Connor, 2012) and other Early Cretaceous sites (You et al., 2006) in northern China, the morphological and taxonomical diversity of the basal Ornithuromorpha—which advanced members include all living birds—remained one of the most poorly understood chapters of avian evolutionary history. The global record of these Cretaceous birds was sparse and limited to either incompletely known taxa such as Ambiortus dementjevi (Kurochkin, 1982), Vorona berivotrensis (Forster et al., 1996), or Ichthyornis dispar (Marsh, 1880; Clarke, 2004), or highly specialized forms such as the flightless Patagopteryx deferrariisi (Alvarenga & Bonaparte, 1992; Chiappe, 2002) and the foot-propelled diving hesperornithiforms (Marsh, 1880; Martin & Tate, 1976). The abundant discoveries from the Early Cretaceous Jehol Group of northern China have resulted in the recognition of a diversity of basal ornithuromorph taxa (e.g., Yanornis martini, Yixianornis grabaui, Gansus yumenensis, Hongshanornis longicresta, Archaeorhynchus spathula, Longicrusavis houi, Parahongshanornis chaoyangensis, Jianchangornis microdonta, Schizooura lii, Piscivoravis lii) (Zhou & Zhang, 2001; Zhou & Zhang, 2005; Zhou & Zhang, 2006; Clarke, Zhou & Zhang, 2006; You et al., 2006; Zhou, Zhang & Li, 2009; O’Connor, Gao & Chiappe, 2010; Li et al., 2010; Zhou, Zhou & O’Connor, 2012; Zhou, Zhou & O’Connor, 2013a), which are known by nearly complete specimens (albeit for the most part preserved bi-dimensionally) and in some instances by multiple specimens (e.g., Yanornis martini, Gansus yumenensis, Archaeorhynchus spathula). These specimens have significantly helped to refine our understanding of basal ornithuromorph morphology but despite these recent advances, specimens of these birds remain relatively rare, the vast majority of the Chinese Early Cretaceous birds belong to more primitive groups such as Enantiornithes and basal pygostylians (e.g., Confuciusornithidae, Sapeornithidae).

Among the newly unearthed avifauna of Chinese basal ornithuromorphs are the hongshanornithids, which are particularly noticeable because of their small size (Table 1) and because they are the only lineage of basal ornithuromorphs known from both the Yixian and Jiufotang formations of the Jehol Group. Originally recognized by the discovery of the holotype of Hongshanornis longicresta (Zhou & Zhang, 2005), subsequent discoveries have added the closely related species Longicrusavis houi (O’Connor, Gao & Chiappe, 2010) and Parahongshanornis chaoyangensis (Li, Zhou & Clarke, 2011). To date, while two additional specimens of Hongshanornis longicresta have been recognized in the literature (Li, Zhou & Clarke, 2011; Zheng et al., 2011), the published anatomical information on this taxon is still limited to the preliminary description of the holotype, which skeleton is entirely preserved as voids on two slabs (Zhou & Zhang, 2005). Here we provide a detailed description of the anatomy of another specimen of Hongshanornis longicresta, DNHM D2945/6 (Figs. 1–4), which preservation provides critical evidence for understanding better the morphology and function of these birds. Furthermore, despite the new discoveries of basal ornithuromorphs, the integument of these birds has remained poorly known in comparison to other basal avians from the Jehol Biota, in particular what refers to the rectricial morphology (Zhou, Zhou & O’Connor, 2012). The complete wing and tail surfaces of DNHM D2945/6, which arguably boasts one of the best-preserved fan-shaped tails from the Mesozoic, reveals important new information about the integumentary evolution and flight capabilities of basal ornithuromorphs.

Table 1 Comparisons of selective measurements in hongshanornithids.

The values highlight the similar sizes and proportions of the three named species of hongshanornithids: Hongshanornis longicresta (DNHM D2945/6 and IVPP V14533), Parahongshanornis chaoyangensis (PMOL-AB00161), and Longicrusavis houi (PKUP 1069).

Element	DNHM D2945/6	IVPP V14533	PMOL-AB00161	PKUP 1069	
Skull	30.5	—	—	30.7	
Humerus	24.6	26.0	29.0	26.0	
Ulna	24.5	24.0	27.0	25.0	
Carpometacarpus	13.0	13.0	12.3	13.1	
Manual digit II		—	18.1	17.7	
Femur	22.0	22.0	24.0	24.3	
Tibiotarsus	35.5	38.0	38.0	37.6	
Tarsometatarsus	20.6	22.0	22.0	21.5	
Pedal digit III	20.0	—	—	—	

Figure 1 Photograph of DNHM D2945.

Red and white arrows point at the maximum extent of the primary and secondary feathers of the wing.

Figure 2 Interpretive drawing of DNHM D2945.

Abbreviations: ce, cervical vertebrae; co, coracoid; fe, femur; fi, fibula; fu, furcula; gl, gastroliths; hu, humerus; il, ilium; mc, metacarpals; mt, metatarsals; pt, proximal tarsals; pu, pubis; ra, radius; rb, ribs; re, rectrices; sc, scapula; sy, synsacrum; ti, tibia; ul, ulna; I–IV, digits (manual or pedal) I–IV. Numbers in inset (close up of gastrolith cluster) refer to those in Table 2.

Figure 3 Photograph of DNHM D2946.

Red and white arrows point at the maximum extent of the primary and secondary feathers of the wing.

Figure 4 Interpretive drawing of DNHM D2946.

Abbreviations: ce, cervical vertebrae; co, coracoid; fe, femur; fi, fibula; fu, furcula; gl, gastroliths; hu, humerus; il, ilium; mc, metacarpals; mt, metatarsals; pu, pubis; ra, radius; rb, ribs; re, rectrices; sc, scapula; st, sternum; sy, synsacrum; ti, tibia; ul, ulna; I–IV, digits (manual or pedal) I–IV. Numbers in inset (close up of gastrolith cluster) refer to those in Table 2.

Materials and Methods

Taphonomy and preservation

DNHM D2945/6 is contained in a slab (DNHM D2945) (Figs. 1 and 2) and counterslab (DNHM D2946) (Figs. 3 and 4), its bones preserved in a semi-three dimensional state and partially split between the two slabs. Such preservation differs from that of the holotypes of Hongshanornis longicresta (IVPP V14533) and Longicrusavis houi (PKUP 1069), which are preserved as voids (even if in a slab and counterslab). Such difference in preservation permits to elucidate morphologies that were obscured in the void-preserved IVPP V14533 and PKUP 1069. DNHM D2945/6 was mechanically prepared by the staff of the Natural History Museum of Los Angeles County, who can verify that the specimen is entirely original.

Locality and stratigraphy

DNHM D2945/6 was collected from the lacustrine deposits of the Yixian Formation at the Dawangzhangzi locality near Lingyuan (Liaoning Province, China), the same locality that yielded the holotype (PKUP 1069) of the closely related Longicrusavis houi (O’Connor, Gao & Chiappe, 2010). The holotype of Hongshanornis longicresta (IVPP V14533) comes from the Shifo locality near Ningcheng (Inner Mongolia, China) and also from the Yixian Formation (Zhou & Zhang, 2005), and is separated from the Dawangzhangzi locality by a distance of approximately 60 km. The stratigraphic and geographic location of other specimens reported as belonging to Hongshanornis longicresta has not been provided in the publications that make reference to these specimens (Li, Zhou & Clarke, 2011; Zheng et al., 2011). The holotype of Parahongshanornis chaoyangensis (PMOL-AB00161) is from the younger Jiufotang Formation in Yuanjiawa Town, Chaoyang City (Liaoning Province, China) (Li, Zhou & Clarke, 2011), thus this specimen is 3 to 5 million years younger (see Swisher et al., 2002; Yang, Li & Jiang, 2007; Chang et al., 2009) than the Yixian hongshanornithids and separated by approximately 100 km from either Dawangzhangzi or Shifo.

Aerodynamic parameters

The aerodynamic parameters and definitions used here follow Pennycuick (2008). Accordingly, wingspan (B) is defined as the distance from one wing tip to the other, with the wings fully stretched out to the sides. Wing area (Swing) is the area of both wings as they are laid on a flat surface, and including the part of the body between the wings. Aspect ratio is the ratio of the wingspan to the mean chord (i.e., the distance from the leading edge to the trailing edge, measured along the direction of the airflow).

Body mass estimates for DNHM D2945/6, based on femoral length, vary substantially according to the different equations provided by Maloiy et al. (1979), Alexander (1983; as listed by Hone et al. (2008)), and Peters & Peters (2009). Maloiy et al.’s (1979) equation results in an unrealistically low body mass, approximately 12 g, comparable to that of many warblers (Dunning, 2008). The equations of Alexander (1983) and Peters & Peters (2009) result in more likely estimates, 50 g and 64 g, respectively, more comparable to the body mass of some cowbirds (e.g., Molothrus bonariensis, Molothrus rufoaxillaris) and orioles (e.g., Icterus chrysater, Icterus gularis) (Dunning, 2008). The mean value of these three estimates is 42 g.

Institutional abbreviations

DNHM, Dalian Natural History Museum, Dalian, China; PKUP, Peking University, Beijing, China; IVPP, Institute of Vertebrate Paleontology and Paleoanthropology, Beijing, China; STM, Tianyu Museum of Nature, Tianyu, China.

Results

Description

In this description we primarily highlight morphological information from the new specimen (DNHM D2945/6) that either supplements or contradicts the description of the holotype (Zhou & Zhang, 2005). Anatomical nomenclature mainly follows Baumel & Witmer (1993); certain structures not cited therein follow Howard (1929). While the Latin terminology used by Baumel & Witmer (1993) is retained for muscles and ligaments, osteological structures are described using the English equivalents of the Latin terms.

The entire skeleton of DNHM D2945/6 is preserved and visible in two slabs; the bones are broken between the two, leaving clear voids. DNHM D2945 preserves the bones of the skull, most of the axial skeleton, right radius, major digit and tibiotarsus, and both feet (Figs. 1 and 2). DNHM D2946 preserves most of the bones of the wings and thoracic girdle, both femora, the left tibiotarsus, and the distal ends of the pubes (Figs. 3 and 4).

Skull

The skull is rather poorly preserved, embedded in a well-indurated rusty concretion; the shape of the skull is well preserved, but few anatomical details can be discerned (Fig. 5). As in the holotype, the skull constricts abruptly into a low and pointed rostrum, different from the more robust rostrum of Longicrusavis houi (O’Connor, Gao & Chiappe, 2010). Impressions of five teeth can be clearly identified in the maxilla (Fig. 5), contradicting the previous description of Hongshanornis longicresta as edentulous (Zhou & Zhang, 2005; Zheng et al., 2011). Teeth were suggested for hongshanornithids on the basis of structures that appeared to be alveoli preserved in the upper jaw of Longicrusavis houi (O’Connor, Gao & Chiappe, 2010). The presence of teeth in the premaxilla, however, cannot be determined in DNHM D2945. There are no obvious pits or scaring on the maxilla or premaxilla to indicate that a beak was present, as in the edentulous Archaeorhynchus spathula (Zhou & Zhang, 2006; Zhou, Zhou & O’Connor, 2013b).

Figure 5 Close up photograph of the skull of DNHM D2945 in right lateral view and detail (inset) of the central portion of its maxilla.

Abbreviations: al, alveoli.

The mandibular symphysial ossification (predentary bone of Zhou & Martin, 2011) is visible as a void in DNHM D2945. The fragmentary remains of two to four teeth are preserved set in the dentary. This confirms the presence of teeth in the dentary of hongshanornithids, which was hypothesized by O’Connor, Gao & Chiappe (2010). Crowns cannot be discerned, but the dentary teeth appear to be smaller than those preserved in the maxilla. The dentary is caudally unforked and unfused to the surangular, as in other basal ornithuromorphs.

Axial skeleton

The cervical vertebrae are poorly preserved, revealing little more than the remnants of low neural spines. The anterior dorsal vertebrae (visible in ventral view in DNHM D2945) bear compressed centra lacking ventral processes; the posterior dorsals have thicker centra. The articulations are amphyplatan or amphicoelic and the elements are not fused in a notarium. The lateral surfaces of the dorsal vertebrae are excavated by a broad fossa, as in other basal ornithuromorphs (e.g., Yanornis martini, Longicrusavis houi).

The synsacrum is incomplete and broken between the two slabs (dorsally exposed in DNHM D2946 and ventrally exposed in DNHM D2945) (Figs. 1–4). A number of well-differentiated costal processes project from the side of the synsacrum—these processes have expanded distal ends for their attachment to the ilium. Judging by the number of costal processes articulated to the ilium, the synsacrum was composed of no less than nine vertebrae (the synsacral count cannot be determined in either the holotype of Hongshanornis longicresta or Longicrusavis houi), which is comparable to other basal ornithuromorphs and more than is typical of the more primitive enantiornithines (Chiappe, 1996). The ventral surface of the synsacrum (anterior half preserved in DNHM D2945) is smooth, lacking a distinct groove (e.g., Archaeorhynchus spathula, Patagopteryx deferrariisi). Dorsally, the portion of synsacrum preserved in slab DNHM D2946—corresponding largely to the postacetabular portion—is longitudinally scarred by a pair of shallow grooves.

At least four uncinate processes can be discerned, including one that is completely preserved (DNHM D2946). These ossifications are long (extending across two ribs) with broad bases and tapered outlines (Fig. 6). The uncinate processes appear not to be fused to the ribs, as in the holotype. Several ventral ribs are preserved in articulation with the sternum (DNHM D2946). The proximal ends of the thoracic ribs are very robust, the expanded proximal portion abruptly narrows towards the much thinner shaft (Fig. 6).

Figure 6 Photograph (A) and interpretive drawing (B) of the thoracic girdle and vertebral series, rib cage, and humeri of DNHM D2946.

Abbreviations: co, coracoid; fu, furcula; hu, humerus; lt, lateral trabecular; ra, radius; rb, ribs; sc, scapula; sp, supracondylar process; st, sternum; up, uncinate process.

Appendicular skeleton

The furcula is delicate and U-shaped as in many other basal ornithuromorpha (e.g., Clarke, Zhou & Zhang, 2006; You et al., 2006; Zhou, Zhang & Li, 2009; Zhou, Zhou & O’Connor, 2013b). The interclavicular angle is estimated to be approximately 45°(Fig. 6). The rami are transversely compressed proximally, becoming more dorsoventrally compressed and wider towards the symphysis. The caudal surface of the bone exhibits a distinct trough running along the distal half of the rami and converging towards the symphysial region. The proximal compression and caudal groove of the furcula of DNHM D2946 are comparable to that reported by Li, Wang & Hou (2011) as diagnostic characters of Parahongshanornis chaoyangensis. The presence of these conditions in DNHM D2946 indicates that such characters are unlikely to be diagnostic of the latter taxon. A cross section of the furcula in slab DNHM D2945 indicates the bone may have been hollow. No long hypocleidium like that reported in the holotype of Hongshanornis longicresta (IVPP V14533) is visible but the symphysial region of DNHM D2945/46 is covered by the distal portion of the left coracoid, thus making it unclear if a hypocleidium was present or not. The similarity between the furcula of DNHM D2945/46 and that of Longicrusavis houi and Parahongshanornis chaoyangensis, taxa lacking a long hypocleidium and possessing only a small tubercle at the symphysis (O’Connor, Gao & Chiappe, 2010; Li, Wang & Hou, 2011), suggests that the furcula of hongshanornithids possibly lacked a well-developed hypocleidium.

The sternum, primarily preserved in DNHM D2946 in dorsal view, has a slightly rounded cranial margin (Fig. 6). Distally on the left side of the sternum, a bony bar with a terminal expansion is preserved. Damage makes it difficult to interpret this region of the sternum. One alternative is that this bar corresponds to a slightly displaced, lateral trabecula with an expanded distal end; the lateral trabecula of the sternum of various basal ornithuromorphs (e.g., Archaeorhynchus spathula, Jianchangornis microdonta, Yixianornis grabaui, Yanornis martini) is distally expanded in varying degrees (Zhou, Zhou & O’Connor, 2013b). Alternatively, it may represent the lateral margin of a sternal fenestra, such as those present in the basal ornithuromorphs Songlingornis linghensis, Yixianornis grabaui, and Yanornis martini (Clarke, Zhou & Zhang, 2006; Zhou, Zhou & O’Connor, 2013b), a feature apparently absent in the sternum of the holotype of Hongshanornis longicresta. Zhou & Zhang (2005) described the sternum of the latter with lateral processes lacking a distal expansion; however the sternum of this specimen is too poorly preserved to confidently support the absence of such expansion.

The strut-like coracoids articulate adjacent to each other on the cranial margin of the sternum, nearly touching each other if not slightly overlapping (Fig. 6). There is a depression (possibly corresponding to the impression of the m. sternocoracoidei of modern birds) on the dorsal surface of the sternal half of these bones (also present in Jehol ornithuromorphs such Jianchangornis microdonta and Yixianornis grabaui). This feature in ornithuromorphs is not as pronounced as the dorsal fossa that excavates the coracoids of many Late Cretaceous enantiornithines (Chiappe & Walker, 2002). The proximal ends of the coracoids are poorly preserved so that a procoracoid process cannot be identified—the presence of this process in Hongshanornis longicresta is not as clear as was suggested by Zhou & Zhang (2005) and remains equivocal in this taxon. However, a procoracoid process is known in almost every other Early Cretaceous ornithuromorph (e.g., Jianchangornis microdonta, Yixianornis grabaui, Yanornis martini, Gansus yumenensis) (Zhou & Zhang, 2001; Clarke, Zhou & Zhang, 2006; You et al., 2006; Zhou, Zhou & O’Connor, 2012). The body of the coracoid exhibits no evidence of either a supracoracoid nerve foramen or medial notch (incisura n. supracoracoidei). Both the lateral and medial borders of this bone are clearly concave, indicating that the convex lateral margin described by Zhou & Zhang (2005) for the poorly preserved holotype is incorrect. A lateral process is present; its squared off morphology is consistent with that of other Early Cretaceous ornithuromorphs (e.g., Ambiortus dementjevi, Yixianornis grabaui, Gansus yumenensis) (Clarke, Zhou & Zhang, 2006; Kurochkin, 1982; You et al., 2006).

Both humeri are exposed in caudal view in DNHM D2946 (Fig. 6). The head is prominent—largely projected caudally—and proximally flat. Such design is more reminiscent to that of the humeral head of Patagopteryx deferrariisi and other basal ornithuromorphs (i.e., Archaeorhynchus spathula, Jianchangornis microdonta) and it differs from the domed-head of modern birds (Chiappe, 2002). The proximal third of the humerus appears to be very broad, expanded as in the holotype of Hongshanornis longicresta, Ichthyornis dispar and Ambiortus dementjevi (Kurochkin, 1985; Clarke, 2004) (Fig. 6). The caudal surface is not perforated by a pneumotricipital foramen and the pneumotricipital fossa is minimally developed. A distinct furrow—presumably the capital incisure—separates the ventral margin of the head from the ventral tubercle. Ventral to the latter, on the ventroproximal corner (and the bicipital area) of the bone, there is a shallow circular depression also present in Ichthyornis dispar, possibly corresponding to the attachment site of the m. pectoralis superficialis (Clarke, 2004). The deltopectoral crest is large—extending longitudinally for more than 1/3 the length of the bone (Fig. 6)—and rounded, lacking the cranial deflection of more advanced ornithuromorphs (neornithines). Distally, the margin of the crest gradually diminishes along the dorsal border of the shaft, typical of Cretaceous ornithuromorphs (e.g., Jianchangornis microdonta, Yixianornis grabaui, Archaeorhynchus spathula), as opposed to the rapid step-like constriction of this crest in most basal birds (e.g., Confuciusornis sanctus, Sapeornis chaoyangensis, Rapaxavis pani) and some basal ornithuromorphs (e.g., Schizooura lii, Zhongjianornis yangi).

In caudal view (DNHM D2946) the distal humeri bear no evidence of humerotriciptial or scapulotricipital grooves (Fig. 6). The olecranon fossa is present but poorly developed. A well-developed dorsal supracondylar process is present, clearest on the right humerus (DNHM D2946) as in Longicrusavis houi (O’Connor, Gao & Chiappe, 2010) and Ichthyornis dispar (Clarke, 2004). The flexor process is small and poorly developed; the transversal distal margin is roughly perpendicular from the longitudinal axis of the shaft as in other ornithuromorphs, not angled as in many enantiornithines (Chiappe & Walker, 2002). The cranial surfaces of the humeri are not visible, planted in the matrix.

The radius is straight and roughly half the width of the ulna. The ulna, comparable in length to that of the humerus (Table 1), is exposed in caudal-dorsal view. Remige papillae are absent. The olecranon process is weakly developed. Distally, the ulna’s dorsal condyle is rounded in caudal view. Near its articulation with the radius, it exhibits a small circular depression that may correspond to the radial depression of living birds (Baumel & Witmer, 1993).

The radius and ulna are in articulation with the proximal carpals and the carpometacarpus (Fig. 7). The radiale is in articulation with the radius. The left carpometacarpus, exposed dorsally in DNHM D2946, is completely fused, proximally and distally. Close to the contact between the major (II) and alular (I) metacarpals there is a raised area. Both the major (II) and minor (III) metacarpals are straight (Fig. 7). Proximally, there is a small depression on the minor metacarpal. The proximal end of the intermetacarpal space extends proximal to the level of the distal end of the alular metacarpal (Fig. 7). The alular (I) metacarpal is subrectangular. An extensor process is not absent, only minimally developed in such a way that the proximal end of the alular metacarpal is slightly more expanded than its distal end (Fig. 7), as in some other Cretaceous ornithurmorphs (e.g., Jianchangornis microdonta, Gansus yumenensis, Yixianornis grabaui).

Figure 7 Photograph of the left manus of DNHM D2945/6 in ventral (A; DNHM D2945) and dorsal (B; DNHM D2946) views.

Abbreviations: mc I-III, metacarpals I-III; p1-I, phalanx 1 (proximal) of digit I (alular digit); p2-I, phalanx 2 (ungual) of digit I (alular digit); p1-II, phalanx 1 (proximal) of digit II (major digit); p2-II, phalanx 2 (intermediate) of digit II (major digit); p3-II, phalanx 3 (ungual) of digit II (major digit); p1-III, phalanx 1 (proximal) of digit III (minor digit); p2-III, phalanx 2 of digit III (minor digit); ra, radius; sl, semilunate carpal; ul, ulna.

The alular digit bears two phalanges. The second phalanx, a claw, extends slightly past the distal end of the major metacarpal, as described by Zhou & Zhang (2005) for the holotype. The major digit has three phalanges; the proximal phalanx is broad bearing a well-developed, sinusoidal lateral flange (Fig. 7). As in the holotype, the intermediate phalanx is S-shaped (Zhou & Zhang, 2005). The claw of the major digit is much smaller than that of the alular digit. The minor digit bears a single, wedge-shaped phalanx that tapers distally; Longicrusavis houi and the Hongshanornis longicresta holotype bear two phalanges on this digit, the second being extremely reduced (and not an ungual) suggesting that this small phalanx is simply not preserved in DNHM D2945/6.

The ilia are poorly preserved and broken between the two slabs; the left is visible in medial view in DNHM D2945, while portions of the right are preserved in DNHM D2946 in dorsal view. The preacetabular wing has a straight dorsal margin that tapers cranially. The caudal half of the ventral margin of the preacetabular wing defines a broad notch, a condition observed in other Jehol ornithuromorphs (e.g., Archaeorhynchus spathula, Schizooura lii) (Zhou, Zhou & O’Connor, 2012; Zhou, Zhou & O’Connor, 2013b).

A fragment of the pubis is preserved near the midshaft of the right tibiotarsus in DNHM D2946 (Figs. 1–4 and 8). The distal ends of both pubes are in contact and only slightly disarticulated. They are not fused but their flat medial surfaces form a short, expanded symphysis, although a distinct ‘boot’, with a prominent caudal projection, like that of some enantiornithines is absent (Chiappe & Walker, 2002) (Fig. 8). The presence of this distal pubic expansion, identical to that present in Parahongshanornis chaoyangensis, indicates that unlike what was claimed by Li, Wang & Hou (2011), this feature is not diagnostic of the latter species. The cross-section of the shaft of the distal portion of the pubis is oval, with the main axis oriented craniocaudally. No pygostyle or caudal vertebrae are preserved in DNHM D2945/6.

Figure 8 Close up photograph of the right knee (medial view) and pubic symphysis (left laterocaudal view) of DNHM D2946.

Note the detail of the gastroliths. Abbreviations: fe, femur; fi, fibula; gl, gastroliths, pu, pubis; ti, tibia. (l) and (r) refer to the left and right pubis, respectively.

The elongate hindlimbs are completely preserved in both slabs as partly bone and partly void (Figs. 1–4). The left femur is preserved in articulation with the left ilium in DNHM D2945—its rounded head is visible through the acetabulum, which is exposed medially. The femoral shaft is only slightly bowed craniocaudally. Its laterodistal end is exposed in DNHM D2946; similar to other Jehol ornithuromorphs (e.g., Yixianornis grabaui), there is minimal development of the fibular trochlea and tibiofibular crest, which are developed in Patagopteryx deferrariisi and ornithurines (Chiappe, 2002; Clarke, 2004).

The tibiotarsus is more than 150% the length of the femur (tibiotarsus: femur = 1.6) (Table 1). The right element is in articulation with the fibula, which is exposed caudally in DNHM D2946. Proximally, the tibiotarsus exhibits a large, well-developed cnemial crest (Fig. 9). This crest, exposed laterally in DNHM D2946, is limited proximally, and projects proximally beyond the proximal articular surface of the tibiotarsus. Its cranial edge develops into an inflated quadrangular prominence that projects laterally (Fig. 9). The morphology of this crest is comparable to that of other Early Cretaceous ornithuromorphs (e.g., Schizooura lii, Yixianornis grabaui) (Clarke, Zhou & Zhang, 2006; Zhou, Zhou & O’Connor, 2012). This cranial prominence is separated by a lateral trough from the proximal articular surface of the tibiotarsus. It is not possible to determine whether the tibiotarsus had one or two cnemial crests; if a cranial cnemial crest were present, it would be embedded in sediment and obstructed by the lateral cnemial crest. In caudal view, the proximal fourth of the tibiotarsus is marked by a robust ridge that slants towards the proximomedial corner of the bone—this feature is not present in Longicrusavis houi, further distinguishing this taxon from Hongshanornis longicresta.

Figure 9 Close up photograph of the left knee (lateral view) of DNHM D2946.

Abbreviations: cn, cnemial crest; fc, fibular condyle; fe, femur; fi, fibula; ld, lateral articular facet of tibia; md, medial articular facet of tibia; ti, tibia.

Distally, the tibia is largely fused to the distal tarsals, however the suture of an ample ascending process of the astragalus remains visible on both the left (DNHM D2946) and right (DNHM D2945) tibiotarsi. The distal condyles are preserved in cranial view in DNHM D2945. The lateral condyle is larger than the medial condyle, as in other basal ornithuromorphs (Chiappe, 1996; Zhou & Zhang, 2006), and separated by a wide intercondylar groove (incisura intercondylaris). The cranial surface of the distal end is scarred by a deep extensor sulcus, which ends near a raised area just proximal to the intercondylar groove—a supratendinal bridge like that of more advanced ornithurines is not developed. The lateral epicondyle is minimally developed and there is only a slightly developed, crescent-like lateral epicondylar depression. Caudally, neither the lateral nor medial crests of the cartilaginous trochlea of the tibia (trochlea cartilaginous tibialis) are developed. These crests are slightly developed in Yixianornis grabaui and well developed in Apsaravis ukhaana (Norell & Clark, 2001; Clarke, Zhou & Zhang, 2006). The fibula is very slender (Figs. 8 and 9)—its distal end does not seem to extend beyond the midpoint of the tibiotarsus.

Both tarsometatarsi are well-exposed in cranial view in DNHM D2945. As reported in the holotype of Hongshanornis longicresta (Zhou & Zhang, 2005), metatarsals II–IV are completely fused to one another. The intercotylar prominence is at best minimally developed. The proximal cotyla are slightly concave—the lateral one is slightly more distally placed than the medial one. Metatarsal III is plantarly displaced so that proximally, metatarsal II and metatarsal IV form ridges defining a recess excavating the central portion of the tarsometatarsus. Such morphology is consistent with that of other basal ornithuromorphs (e.g., Yanornis martini, Yixianornis grabaui, Gansus yumenensis, Ichthyornis dispar). Inside this proximocentral recess there is a foramen located between metatarsals III and IV, and medial to it, a tubercle on metatarsal II (possibly corresponding to the m. tibialis cranialis tuberosity of modern birds). Metatarsal III is the longest; metatarsal IV is slightly shorter, followed by the even shorter metatarsal II (Fig. 10). Metatarsal I is robust and fairly straight, with a concave medial margin. A distal vascular foramen is located between metatarsals III and IV as in Longicrusavis houi; the distal margin of the foramen is raised. The foramen seems to penetrate the bone at an oblique angle—from cranial to plantar surfaces. All trochleae appear to be ginglymous.

Figure 10 Close up photograph of the feet and rectrices of DNHM D2945.

Abbreviations: mt, metatarsals; re, rectrices; I-IV, digits I-IV.

Digit III is the longest (Table 1); digit II is substantially shorter than IV (Fig. 10). Digit I is short and slender. All the pedal phalanges are long and slender and decrease in length distally. The phalanges of the second and third digits are approximately 2/3 the length of the preceding phalanx. The phalanges of digit IV are subequal, but still slightly decrease in length distally. The ungual phalanges bear distinct flexor tubercles. The morphology of the distal tarsometatarsus and the proportions of the pedal phalanges are consistent with cursorial function (Hopson, 2001), as in other Early Cretaceous ornithuromorphs.

Plumage

DNHM D2945/6 preserves significant portions of the plumage of the wing, tail, and around the skull and neck. The feathers preserved over the skull question the assumption that this species was characterized by the presence of a feathery crest projecting from the head (Zhou & Zhang, 2005). DNHM D2945/6 shows nothing of that sort. The relatively short feathers of the head become gradually shorter until the plumage ends at the junction of the rostrum and the orbit, thus indicating that the culmen was devoid of feathers (whether it was covered by skin or a corneous beak is unknown). Furthermore, contrary to what was suggested by Zheng et al. (2011) for another specimen putatively identified as Hongshanornis longicresta (STM 35-3), DNHM D2945/6 indicates that these birds had relatively broad, but long and tapered, wings (Figs. 1–4 and 11).

Figure 11 Interpretive drawing of the outline of the wings and feathered tail of DNHM D2945/6 (see Figs. 1 and 3 for the maximum extent of the primaries and secondaries).

The aerodynamic parameters used in this study were derived from this reconstruction (see Materials and Methods).

The most distal primaries are clearly much shorter than the remaining primaries. Based on the well-preserved outline of the wing, our estimation of the wingspan of DNHM D2945/6 is approximately 0.32 m and the wing area is 0.016 m2 (Fig. 11). There is exquisite evidence of rectrices forming a fan-like tail preserved in natural orientation, about the extended feet. Although visualization of individual feathers is difficult due to overlap, the tail appears to be composed of at least 10 vaned rectrices. The precise degree of asymmetry of these feathers cannot be ascertained but it is clear that none of them were strongly asymmetric (Fig. 10). The caudal margin of the feathered tail is rounded and the pair of central rectrices projects distally more than the feathers on either side, thus suggesting a gently graded tail (comparable to the one described for Piscivoravis lii; see Zhou, Zhou & O’Connor, 2013a). The holotype of Hongshanornis longicresta also shows evidence of an extensive feathered tail as a series of partially preserved vaned feathers between the feet (IVPP V14533A); these are very faint and were not described in the original description (see O’Connor, Gao & Chiappe, 2010). In the holotype of Hongshanornis longicresta, the tail is clearly incomplete and only four feathers can be discerned; the tail in DNHM D2945/6 appears complete. The number of preserved feathers is thus greater than the one described in the younger ornithuromorphs Yixianornis grabaui (minimum of eight described) and Piscivoravis lii (at least six described) from the Jiufotang Formation (Clarke, Zhou & Zhang, 2006; Zhou, Zhou & O’Connor, 2013a).

Gastroliths

At least 11 small geo-gastroliths are clustered in the visceral and pelvic region of DNHM D2945/6. Some of these ‘stomach stones’ are still in place while others are represented by the voids left on the slabs (Figs. 1–4 and 8). The smallest of the stones is nearly half the size the largest (1.71 mm to 3.08 mm) with the mean being 2.2 mm (Table 2). They range in shape from subspherical to oblong. It is unclear whether gastroliths were preserved with the holotype (IVPP V14533A), although more than 50 small (∼1 mm) geo-gastroliths were reported in the poorly preserved STM 35-3, a specimen identified as of Hongshanornis longicresta by Zheng et al. (2011). Despite the fact that Zheng et al. (2011) did not provide any anatomical evidence supporting the identification of STM 35-5 as Hongshanornis longicresta, the discovery of DNHM D2945/6 provides unquestionable evidence of the behavior of ingesting grit by this basal ornithuromorph species.

Table 2 Measurements (mm) of the gastroliths of DNHM D2945/6.

Left column numbers correspond to numbers on Figs. 2 and 4.

	D2945	D2946	Mean value	
1	2.21	2.27	2.24	
2	3.06	3.09	3.08	
3	2.31	2.07	2.19	
4	2.05	2.15	2.10	
5	2.15	2.11	2.13	
6	2.90	2.59	2.75	
7	1.84	1.76	1.80	
8	1.76	1.79	1.78	
9	1.74	2.08	1.91	
10	1.84	1.56	1.70	
11	2.51	3.13	2.82	

Discussion

The specimen described here, DNHM D2945/6, is indistinguishable from the holotype of Hongshanornis longicresta and is thus referred to this taxon. DNHM D2945/6 is from the same locality and formation (Yixian) that yielded the closely related Longicrusavis houi (PKVP 1069; O’Connor, Gao & Chiappe, 2010). The coexistence of these two closely related taxa in a single fauna suggests that Hongshanornithidae was a diverse component of the avifauna of the Yixian Formation.

DNHM D2945/6 reveals important new information regarding the morphology, diet, and ecology of Hongshanornis longicresta and other Early Cretaceous ornithuromorphs. The new specimen of Hongshanornis longicresta preserves actual fossilized bone rather than voids of bone, as in the holotype (IVPP V 14533), and as such this new specimen helps clarify the osteology of the taxon. DNHM D2945/6 documents the presence of two different skull morphologies within Hongshanornithidae. It also confirms that the upper and lower jaws of these birds were toothed, a conclusion previously suggested by morphologies preserved in the holotype of Longicrusavis houi (O’Connor, Gao & Chiappe, 2010). Additionally, DNHM D2945/6 shows that the lateral margin of the coracoid of hongshanornithids was concave, as is typical of other basal ornithuromorphs, and not convex as reported by Zhou & Zhang (2005). DNHM D2945/6 also preserves a dorsal supracondylar process, a feature present in Longicrusavis houi and many extant shorebirds.

The presence of a hypocleidium in Hongshanornis longicresta is still controversial. This process does not appear present in DNHM D2945/6, however preservation prevents us from making an unequivocal statement. While most known Early Cretaceous ornithuromorphs do not possess even the smallest hypocleidium, basal ornithuromorphs with furculae bearing a distinct hypocleidium have been recently reported (Zhou, Zhou & O’Connor, 2012). Such discoveries lend credibility to Zhou and Zhang’s (2005) claim that the furcula of Hongshanornis longicresta possessed a hypocleidium, although well-preserved specimens are necessary to corroborate such a claim.

Furthermore, the anatomy of DNHM D2945/6 partially undermines the anatomical basis used by Li, Wang & Hou (2011) to diagnose the hongshanornithid Parahongshanornis chaoyangensis. The compression of the proximal portion of the furcula and caudal groove of this bone in DNHM D2945/6 indicate that these characters cannot be considered as diagnostic of Parahongshanornis chaoyangensis. This is the same for the distal expansion of the pubes, a condition that is identical in DNHM D2945/6 and the holotype of Parahongshanornis chaoyangensis.

Diet

The presence of teeth on both the maxilla and the dentary, and the reinterpretation of the hongshanornithids as toothed birds, impact earlier ecological inferences for the clade, suggesting they were less specialized trophically than previously imagined. The presence of at least two distinct skull morphologies within Hongshanornithidae indicates intraclade trophic diversity, and niche partitioning within the specialized wading ecology inferred to have been occupied by this clade.

The geo-gastroliths preserved associated with the visceral region of DNHM D2945/6 offer a glimpse into the digestive system and dietary preferences of hongshanornithids. A variety of aquatic and terrestrial organisms regularly ingest sand, fine gravel, or coarse sand. These ‘stomach stones’ or grit are assumed to perform a variety of functions ranging from acting like ballast in aquatic animals to parasite control and hunger placation (Wings, 2007). Most typically, however, geo-gastroliths are interpreted as grinding devices assisting the digestion of hard food items. Geo-gastroliths are not only common among living crocodiles (Taylor, 1993; Henderson, 2003) and birds (Gionfriddo & Best, 1999) but they also have been documented in a variety of extinct archosaurian clades, including pterosaurs (Codorniú, Chiappe & Cid, 2013) and every major group of dinosaurs (Osborn, 1924; Ji et al., 1998; Weems, Culp & Wings, 2007; Wings & Sander, 2007). Geo-gastroliths have also been discovered in association with the skeletons of a number of Early Cretaceous ornithuromorphs from China (e.g., Yanornis martini, Gansus yumenensis, Archaeorhynchus spathula) (Zhou et al., 2004; You et al., 2006; Zhou, Zhou & O’Connor, 2013b). The presence of geo-gastroliths in fossil dinosaurs (including birds) is generally regarded as indicative of herbivory (Ji et al., 1998; Gionfriddo & Best, 1999; Kobayashi et al., 1999; Wings, 2007; Xu et al., 2009; Zanno & Makovicky, 2011; Brusatte, 2012), however such a correlation is not as consistent as often assumed. In the case of DNHM D2945/6, the closely associated cluster of fairly evenly sized stones located in the abdominal cavity distal to the sternum but proximal to the pelvic girdle, is consistent with the interpretation of these geo-gastroliths as gizzard stones. Furthermore, the report by Zheng et al. (2011) of a specimen (STM 35-3, identified as of Hongshanornis longicresta) containing seeds in a crop provides evidence that these birds were granivorous. Such evidence is consistent with both the preservation of geo-gastroliths in DNHM D2945/6 and STM 35-3, and the functional interpretation that highlights the role these stones play in processing hard foods.

Caudal plumage

Very little is known about the rectrices of basal ornithuromorphs and DNHM D2945/6 represents what is possibly the most informative example for understanding the morphology of the tail of these birds. Only two types of feathered tails—fan-shaped and forked—have been reported for basal ornithuromorphs (Zhou & Zhang, 2001; Clarke, Zhou & Zhang, 2006; Zhou, Zhou & O’Connor, 2012; Zhou, Zhou & O’Connor, 2013a). Such limited diversity contrasts with what is known for their sister-group, the Enantiornithes, in which a larger number of tail morphologies (including one inferred to be aerodynamic; O’Connor et al., 2009) are known (Zhang & Zhou, 2000; Zheng, Zhang & Hou, 2007; O’Connor, Gao & Chiappe, 2010; O’Connor et al., 2012).

DNHM D2945/6 provides evidence that as early as 125 million year ago, basal ornithuromorphs had evolved a feathered tail comparable in size and design to those of many modern birds. Such tail had rectrices suitable of fanning and capable of generating aerodynamic forces much greater than those of more basal birds (Gatesy & Dial, 1996).

In living birds, all but the central pair of rectrices of a fan-shaped tail anchor on a musculo-adipose organ, the rectricial bulb, which controls the fanning of the feathers and as such, plays a critical role in flight (Gatesy & Dial, 1996). The rectricial bulb is in turn supported by, and intimately connected to, the pygostyle. Nonetheless, because no direct evidence of a rectricial bulb has ever been found in any Mesozoic bird, whether the pygostyle is a reliable osteological correlate of the rectricial bulb has been questioned (Clarke, Zhou & Zhang, 2006). The morphology of the pygostyle of basal ornithuromorphs (and of their living relatives) differs from that typical of more basal birds, which commonly possess a long, robust, and club-shaped pygostyle (e.g., Confuciusornithidae, Enantiornithes). In contrast, the pygostyle of ornithuromorphs is relatively small, more delicate, and plough-shaped—even if the pygostyle of hongshanornithids is poorly known (and missing in DNHM 2945/6), its morphology seems to agree with that of other basal ornithuromorphs (Zhou & Zhang, 2005). The morphological difference between the pygostyle of ornithuromorphs and that of more basal birds led Clarke, Zhou & Zhang (2006) to hypothesize that the rectricial bulb evolved in concert with both the modern avian pygostyle (plough-shaped) and aerodynamic tail morphologies such as those known for basal ornithuromorph taxa. This hypothesis, however, has been somewhat challenged by the recent report of a tail formed by long, shafted rectrices in the enantiornithine Shanweiniao cooperorum (O’Connor et al., 2009). In the holotype and only known specimen of this taxon, the impressions of four vaned feathers can be seen projecting from the end of the caudal vertebral series. Such evidence suggests that the presence of a large and club-shaped pygostyle, like the one characteristic of enantiornithines, might not have excluded the development of a fan-shaped feather tail among non-ornithuromorph birds from such tail feather morphologies (O’Connor et al., 2009), and therefore that a rectricial bulb might have evolved either prior to the origin of the Ornithuromorpha or independently in more than one clade of birds (e.g., Ornithuromorpha and Enantiornithes). The derived phylogenetic position of Shanweiniao cooperorum, nested among advanced enantiornithines (O’Connor & Zhou, 2012), and the widespread presence of pintail morphologies lacking vaned rectrices capable of fanning among other enantiornithines (O’Connor et al., 2012) favor interpretations of the aerodynamic tail of Shanweiniao (and presumably the rectricial bulb that operated the fanning of its rectrices) as an independent evolutionary event. Likewise, the presence of fan-shaped feathered tails and modern-like pygostyles in a variety of basal ornithuromorphs (e.g., Hongshanornis longicresta, Yixianornis grabaui, Yanornis martini, Piscivoravis lii) supports the argument that such morphologies are ancestral for Ornithuromorpha (Clarke, Zhou & Zhang, 2006).

Aerodynamics

DNHM D2945/6 possesses relatively broad, but tapered wings. Compared to a general regression of mass against wing area for birds (data taken from Greenewalt, 1962), the relative wing area for DNHM D2945/6 is somewhat larger than average, when using the mean body mass estimate (42 g; residual of −0.37) (see Materials and Methods). However, using the heaviest mass estimate for DNHM D2945/6 (65 g) (see Materials and Methods) yields an average wing area, and therefore a typical wing loading compared to a modern bird of similar size (residual of −0.04).

The overall size of DNHM D2945/6, along with its wing shape, is qualitatively similar to that of some living passerines with flexible diets (e.g., Monticola solitarius, Sturnus vulgaris, Turdus naumanni). These same living birds also tend to have somewhat tapering wings without extensive wing tip slotting. As a result, we tentatively suggest that DNHM D2945/6 may have lacked extensive wingtip slotting, even if direct evidence of this is not available in the specimen. Additional specimens will be required to confirm this prediction. Tip slots are most effective at slow speeds, where they can increase effective aspect ratio in wings with low geometric aspect ratio (Tucker, 1993). As a result, it is generally expected that birds without tip slots tend to either have high geometric aspect ratio (which is not relevant to DNHM D2945/6), or else tend to fly at relatively high speeds. Based on its intermediate size and moderate aspect ratio, DNHM D2945/6 would be expected to flap continuously at low speeds, but may have switched to flap-bounding (a cycle of flapping and bounding with the wings flexed) at higher rates of travel, since this type of gait transition is seen among modern birds with similar wing shape and body size to DNHM D2945/6. We note that such gait transitions are related to flight speed and cost of transport, both of which relate to total body size and wing loading (Tobalske, Peacock & Dial, 1999; Tobalske, 2001). Since we have reliable measurements of the specimen dimensions, conservative predictions of gait are possible for DNHM D2945/6, even though the wings of the specimen lack feather details. For the same reasons, we do not expect that these gait transitions would be particularly sensitive to specifics of osteology (i.e., differences in skeletal structure between basal ornithuromorphs and modern birds). So long as DNHM D2945/6 was able to fly continuously for significant distances, the same basic patterns of cost of transport relative to size and flight speed should apply to both living neornithines and DNHM D2945/6.

Tail positioning and effective functional area are more difficult to estimate in fossil taxa than the same variables for the wings, because avian tails can be used at extremely fanned or collapsed states (or any number of positions in between). While the degree of asymmetry of the rectrices is unclear, the lateral feathers of the tail in DNHM D2945/6 appear to be less asymmetrical than it would expected if these feathers were regularly oriented with their long axis fully transverse to the air flow (as would be the case for primary feathers on the wings or the lateral rectrices in a fully fanned tail). This suggests that the tail was typically deployed as a partially fanned “wedge”, rather than a full fan. This is not uncommon among modern birds with long tails, such as flycatchers, accipiters, and sunbitterns (M Habib, pers. observation).

The preserved plumage of DNHM D2945/6 indicates that hongshanornithids had wing and tail surfaces comparable to those of living birds of similar sizes. In general, the morphology of the wing and feathered tail of DNHM D2945/6, combined with estimates of its weight, is indicative of a flight mode comparable to the intermittent flight of many medium-sized passerines and congruent with the conclusions reached by Close & Rayfield (2012), whose geometric morphometric analysis of furculae interpreted Hongshanornis longicresta as a continuous flapper.

We thank Doug Goodreau for preparing the specimen, Stephanie Abramowicz for creating the illustrations, and Maureen Walsh for editing the manuscript.

Additional Information and Declarations

Competing Interests

Author Contributions

Luis M. Chiappe and Jesus Marugan-Lobon serve as Academic Editors for PeerJ.

Luis M. Chiappe conceived and designed the experiments, performed the experiments, analyzed the data, wrote the paper.

Zhao Bo, Gao Chunling, Wang Xuri and Meng Qingjin access and curation of specimen.

Jingmai K. O’Connor and Michael Habib performed the experiments, analyzed the data, wrote the paper.

Jesus Marugan-Lobon performed the experiments, analyzed the data.

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
