# Peer review of "A new specimen of the Early Cretaceous bird Hongshanornis longicresta: insights into the aerodynamics and diet of a basal ornithuromorph"

_PeerJ, doi:10.7717/peerj.234_

## Round 0.1 · original submission · Minor Revisions

This paper is clearly worth publishing as this is a well-preserved Mesozoic bird skeleton that elucidates skeletal evolution and clarifies anatomical details. Although it is clear from the photos that this bird had feathers, I agree with Reviewer 1 that more detailed photos of these are necessary to uphold the description of plumage and aerodynamic performance. If such photos cannot be provided, these sections should be deleted. I do believe that the discussion of gastroliths is appropriate in length. If the quality of the teeth allow a better description, this would be useful.

Reviewer 1 ·

Basic reporting

The paper describes another specimen of a mesozoic bird. The osteological description is clearly written and well figured.

The claim that the specimen reveals new information on diet is only partially valid since a crop with seeds has already been published for this species(Zheng et al). The presence of teeth is important but no mention was made of how this impacted their diet. The discussion on gastroliths has already been published in several places. It would be more important to examine microwear or isotopes for the teeth and/or gastroliths. In some way the description of the teeth should indicate why they support granivory.

The authors claim "well-preserved" wing and tail feathers however, only the faint impression of the terminus of the retrices are visible from their photographs. Although there is an apparent outline of the wing, individual feather details cannot be seen in the photographs provided. For instance, there is no area on the fossilized wing where one can determine whether or not the feathers are asymmetrical. This is also true for the purported wing slots -- because of these issues it is difficult to see how an aerodynamic model can be constructed.

Also problematic is that none of the feathers can be traced to their respective attachment sites. Perhaps this is the quality of the photographs. I would suggest cross-lighting or laser/UV to enhance such details.

Experimental design

Because of the lack of details on the feathers and wing the aerodynamics are questionable especially in light of the very broad results reported for mass.

The reconstructed body outline with plumage does not appear accurate. Please check the proportions of the alula to the wing.

The specimen should be examined radiographically to ensure its integrity and to reveal osteologic details such as the shape of the hypocleidiem.

Validity of the findings

The osteological description is well written and clearly adds to our knowledge of this type of bird.

However, the conclusions pertaining to the mode of flight is over reaching and thus the commentary is speculation rather than solid results and should be identified as such. In particular, the Aerodynamic section mentions wing tip slotting in modern birds of this type but there is absolutley no evidence that can seen to demonstrate their specimen had tip slots.

It was not apparent if the analysis would be skewed based on feather geometry which was not mentioned.

Additional comments

The specimen needs to be re-photographed in a way to display the feathers and verify wing slots and feather symmetry.

Although the distal portion of the tail is preserved, the length is an assumption since the feathers cannot be traced to their origin in relation to the specimen.

The discussion on gastroliths provides no new information and should be reduced in length.

The discussion on pygostyle is apparently based on other specimens and it is difficult to evaluate in this paper without the other specimens being figured.

The references contain gray literature and should be removed.

Reviewer 2 ·

Basic reporting

Pass. Italics seem to be missing in the abstract. The introduction seems sufficient and the review, good.

Experimental design

Pass. Clearly organized.

Validity of the findings

Two areas for improvement.

Morphologies of the caudal rectrices as well as the recovered feathered tail in ornithurines are certainly of importance regarding the origin of tail fanning in modern birds. Rather than comparing to basal ornithurines with tail feathering preserved, the authors make most comparisons with a proposed fan-tailed enantiornithine, in which the bony pygostyle is quite different in form.. Additional comparisons of the tail in Hongshanornithidae with other ornithurines (e.g., Yanornis, Yixianornis), which have a similar pygostyle and caudal rectrices could be more relevant and provide further insight into the early evolution of a fan tail.


Regarding the aerodynamic analysis, by reconstructing the wing and tail surface and body weight, the authors concluded a similar flight mode applied by basal ornithurines and living passerines with a similar wing shape and body size. However, this analysis does not take into account the absence of many skeletal features present in living species that are absent in early ornithurines. At present conclusions drawn from these comparisons are interesting but not particularly convincing. Perhaps at least some caveats could be added.

Additional comments

The paper reports an interesting new specimen of basal ornithurine, referable to Hongshanornis longicresta. In addition to generally better preserving skeletal detail in comparison with the holotype, the new specimen has also preserved a clear suit of wing and tail feathers, as well as the gastroliths in the visceral region. Limited known data on basal ornithurines make the reported specimen a valuable record to understand the early evolution of this particular group of birds from Cretaceous. Ecology and aerodynamic performance of the species were further inferred in the paper based on the new morphologies recovered from the specimen.

The authors did a good job in describing the detailed anatomy of the specimen. Several features which poorly exposed in the holotype specimen are illuminated here, e.g., the presence of maxillae and dentary teeth, concave lateral margin of the coracoid, the expanded pubic boot and the cnemial crest on the tibiotarsus. The authors additionally found several proposed diagnostic characters of Parahongshanornis to be actually shared by the new specimen as well, raising the question of the justification of Parahongshanornis as a new species and genus.

---

## Round 0.2 · accepted · Accept

The authors made most of the (minor) changes suggested by the reviewers. They considered other suggestions seriously, and more changes were not possible, mostly due to the preservational status of the specimen.